# Response Surface Methodology for Optimization of Process Parameters and Antioxidant Properties of Mulberry (*Morus alba* L.) Leaves by Extrusion

**DOI:** 10.3390/molecules25225231

**Published:** 2020-11-10

**Authors:** Mina Kim, Dong-Geon Nam, Wan-Taek Ju, Jeong-Sook Choe, Ae-Jin Choi

**Affiliations:** 1Division of Functional Food & Nutrition, Department of Agrofood Resources, National Institute of Agricultural Sciences, Rural Development Administration, Wanju-gun 55365, Korea; lucidminakim@gmail.com (M.K.); realfood@korea.kr (D.-G.N.); swany@korea.kr (J.-S.C.); 2Division of Sericulture & Apiculture, Department of Agricultural Biology, National Institute of Agricultural Sciences, Rural Development Administration, Wanju-gun 55365, Korea; wantaek@korea.kr; 3Planning Team in Korea Program for International Cooperation in Agricultural Technology (KOPIA), Rural Development Administration, Jeonju-si 54875, Korea

**Keywords:** leaf, *Morus alba*, flavonoid, reactive oxygen species, macrophage, apoptosis

## Abstract

Mulberry (*Morus alba* L.) leaves (MLs), originally used to feed silkworms, have recently been recognized as a food ingredient containing health-beneficial, bioactive compounds. In this study, the extrusion process was applied for the enhancement of the amount of extractable flavonoids from MLs. Extrusion conditions were optimized by water solubility index, total phenolic content, and total flavonoid content (TF) using response surface methodology, and antioxidative stress activities were evaluated in macrophage cells. According to the significance of regression coefficients of TF, the optimal extrusion parameters were set as barrel temperature of 114 °C, moisture feed content of 20%, and screw speed of 232 rpm. Under these conditions, the TF of extruded ML reached to 0.91% and improved by 63% compared with raw ML. Fifteen flavonoids were analyzed using ultra-high-performance liquid chromatograph coupled with photodiode array detection and quadrupole time-of-flight mass spectrometry (UPLC-PDA-QTOF/MS), and the extrusion resulted in increases in quercetin-3-gentiobioside, quercetin-3,7-di-*O*-glucoside, kaempferol-3,7-di-*O*-glucoside, rutin, isoquercitrin, and moragrol C. Besides, regarding antioxidative activity, extruded ML water extract inhibited the production of H_2_O_2_-induced reactive oxygen species and attenuated nuclear morphology alterations in macrophage cells. The findings of this study should be useful in food processing design to improve the extractable functional compounds in MLs.

## 1. Introduction

Mulberry (*Morus alba* L.) trees are cultivated mainly in Asian countries, and their leaves are used to feed silkworms [1]. With the recent increase in the production of mulberry fruits, the demand for technologies and mulberry cultivars needed for fruit production has been on the rise [2]. Various varieties of black mulberry trees are being developed for high productivity and large fruit sizes [3]. Recently, mulberry leaves (MLs) have been recognized as edible with medicinal properties owing to their phytochemical content [4]. Phytochemical studies have shown that MLs are a rich source of a variety of constituents, including flavonoids [3,5], polysaccharides [1,6], alkaloids [7], and 1-deoxynojirimycin [8]. Besides, several different flavonol (i.e., quercetin and kaempferol) glycosides have been reported as antioxidant constituents in MLs. These active components contain reactive oxygen species (ROS) scavengers [4,9,10]. Substantial previous evidences indicated that MLs could be a potential functional ingredient for the production of healthy foods [5,6,7].

However, the leaves of black mulberry varieties are generally discarded because they are planted only for fruit, and the leaves are not advisable for use as food even for silkworms [11]. The utilization of MLs has not been considered at the farm-related site, and most ML-related products are in the form of powder or tea [12]. Besides, for the bioactive compounds of plants to be bioaccessible, they have to be released from cell walls and various structural compositions that affect the absorption properties [13]. Because most phenolic compounds are linked by ester bonds, located mainly in cell walls [14], the ester bonds between the phenolic compounds in ML make them resistant to release and extraction [15]. Thus, with consideration to physiologically available bioactive compounds and market expansion, the application of processing technology to MLs to produce various value-added materials is of great research interest [16]. The MLs market needs to become elaborated and diverse as the mulberry fruit market.

Extrusion is a thermomechanical operation in which high pressure and shear forces are used for a short-time, continuous, and stepwise processing [13]. The fiber-rich materials undergo cellulose polymer transformation during extrusion to their physicochemical properties to achieve the degradation of macromolecules [17]. Extrusion of leaves results in the easier release of active compounds from the matrix, due to degradation of the fibrous structure [13]. It consequently changes the functional properties and enhances the bioavailability of nutrients [15,18]. Studies have demonstrated that extrusion can be an alternative processing technique for increasing the yields of effective components, such as saponins, ginsenosides, and flavonoids, as well as improving their biological potential [15,18]. There are some studies on vegetable by-product usage in extrusion [13,15,18]. The majority of previous studies have established the combination of starch ingredients with leaves using extrusion processing for snack or cereal production [19,20,21,22]. There are limited studies in the literature on the enhancement of solubility and extractability of flavonoids from MLs that have been processed by the extrusion technique [4].

Therefore, the objective of the present study was to optimize the effect of different extrusion conditions (i.e., barrel temperature, T; feed moisture content, M; screw speed, S) on the amount of extractable flavonoids from MLs using response surface methodology (RSM), thereby providing a reference for the profile of flavonoids through a qualitative-quantitative approach. In addition, we confirmed the effect of antioxidative properties of an optimal extruded ML water extract using chemical methods and the RAW 264.7 macrophage cell line.

## 2. Results

### 2.1. Response Surface Optimization and Verification of Predictive Model

#### 2.1.1. Determining Levels for Independent Variables

To determine the levels of independent variables, such as T (°C) and M (%), dependent variables (i.e., water solubility index, WSI; total phenolic content, TP; total flavonoid content, TF) were analyzed. A factorial design was derived from these results, and the main effects of independent variables and an interaction plot were obtained (Appendix A). In the regression analysis, R^2^ values of WSI, TP, and TF were 93.32%, 90.83%, and 90.68% respectively, and the predicted R^2^ values were 84.95%, 79.36%, and 79.02%, respectively. WSI, TP, and TF increased as M increased (Appendix A). Considering the interaction of T and M, high levels of WSI and TF were indicated at T of 120 °C and M of 30% (Appendix A), and high levels of TP were indicated at T of 130 °C and M of 30% (Appendix A). Therefore, 110, 120, and 130 °C were chosen for the coded T variable levels at −1, 0, and +1, respectively. In addition, the three design levels of 20%, 30%, and 40% were selected for M.

#### 2.1.2. Statistical Analysis and Model Fitting

The response values for each dependent variable in different experimental combinations are shown in Table 1. Extruded ML showed higher WSI, TP, and TF than raw ML (Run No. 0: non-extruded). Multiple regression analysis was performed, and the response variables based on the results in Table 1 were related by the following second-order polynomial equations.
molecules-25-05231-t001_Table 1Table 1Experimental design using Box-Behnken design and the response values.Run No.Independent VariablesDependent VariablesCoded LevelsActual LevelsResponsesX_1_X_2_X_3_TMSWSI (%)TP (%)TF (%)0------34.54 ± 0.411.04 ± 0.020.65 ± 0.0210001203025039.26 ± 0.101.22 ± 0.031.00 ± 0.032−1−101102025041.88 ± 0.141.37 ± 0.011.09 ± 0.0331101304025041.18 ± 0.291.31 ± 0.010.94 ± 0.0340111204030039.20 ± 0.151.23 ± 0.000.91 ± 0.015−1011103030040.78 ± 0.391.31 ± 0.000.97 ± 0.0260−1−11202020042.06 ± 0.311.40 ± 0.021.10 ± 0.0270001203025040.58 ± 0.211.36 ± 0.021.03 ± 0.0280001203025040.53 ± 0.301.36 ± 0.011.03 ± 0.0191−101302025041.55 ± 0.151.39 ± 0.011.04 ± 0.02100−111202030041.54 ± 0.181.32 ± 0.021.02 ± 0.011110−11303020039.12 ± 0.271.30 ± 0.010.91 ± 0.0512−10−11103020038.97 ± 0.151.29 ± 0.010.93 ± 0.02131011303030040.93 ± 0.321.20 ± 0.140.98 ± 0.0214−1101104025039.57 ± 0.101.23 ± 0.010.83 ± 0.021501−11204020038.78 ± 0.361.24 ± 0.010.81 ± 0.03-, raw mulberry leaf powder. X_1_, T, barrel temperature (°C); X_2_, M, feed moisture content (%); X_3_, S, screw speed (rpm). WSI, water solubility index; TP, total phenolic content; TF, total flavonoid content. Means with different superscripts letters in each column are significantly different (*p* < 0.05).
89.4 − 0.741X_1_ − 1.106X_2_ + 0.0879X_3_ + 0.00256X_1_^2^ + 0.00697X_2_^2^ − 0.000158X_3_^2^ + 0.00487X_1_X_2_(1)
−0.31 + 0.0107X_1_ − 0.0345X_2_ + 0.01312X_3_ − 0.000014X_3_^2^ + 0.000167X_1_X_2_ − 0.000062X_1_X_3_ + 0.000035X_2_X_3_(2)
−2.13 + 0.0615X_1_ − 0.0796X_2_ + 0.00558X_3_ − 0.000304X_1_^2^ − 0.000016X_3_^2^ + 0.000401X_1_X_2_ + 0.000088X_2_X_3_(3)

The results of the statistical tests performed by analysis of variance (ANOVA) are indicated in Table 2. The *p*-value was used to evaluate the significance of each coefficient, which might indicate the pattern of interactions between the variables. WSI, TP, and TF were significantly dependent on M, which had a negative linear effect. In addition, in the case of TF, the positive cross-product coefficients (T × M, *p* < 0.05; M × S, *p* < 0.01) and the negative quadratic term coefficients (T^2^, *p* < 0.05; S^2^, *p* < 0.05) were significant (*p* < 0.05), whereas coefficients of the other term were not significant. In particular, the regression model for TF showing a high R^2^ value (96.18%) indicated good explanatory ability with the fitted model, suggesting that a close correlation was achieved as the R^2^ value became higher than 0.8 [9]. The value of the adjusted R^2^ was also highly significant for TF (R^2^ = 0.9236), and the predicted R^2^ was 0.7725. Lack of fit was also determined, and it was not significant (*p* > 0.05). Finally, TF provided evidence useful for predicting the optimal extrusion conditions.

#### 2.1.3. Optimization of Extrusion Conditions

Three-dimensional (3D) surface plots provide a method to visualize the relationships between responses and experimental levels of each variable [8]. Based on the second-order polynomial equations of TF, which can be used to determine optimal levels of the variables, the results are presented in Figure 1. The response surface plot showed that TF decreased with increasing T and M. The response surface plot of M and S for TF showed that TF increased dramatically with decreasing M (Figure 1). 

For optimization of TF in extruded ML, according to the significance of the regression coefficients of the quadratic polynomial model, the optimal extrusion conditions obtained by RSM were T of 114.44 °C, M of 20%, and S of 231.31 rpm (Appendix A). The value predicted using Minitab software was verified under the optimal extrusion conditions. To facilitate actual production, modified conditions were set as T of 114 °C, M of 20%, and S of 232 rpm. Under these conditions, the mean actual TF% was 0.91 ± 0.04, and the experimental TF% of raw ML was 0.56 ± 0.02. The RSM model, with an adequate extrusion process, improved TF by 62.5%.

### 2.2. Effect of Extrusion on Flavonoids in ML

Prior studies reported that phenolics are present in extruded food, discovering that extrusion could improve the accessibility of phenolics [23]. In the present study, flavonoid compounds were analyzed using ultra-high-performance liquid chromatograph coupled with photodiode array detection and quadrupole time-of-flight mass spectrometry (UPLC/PDA-QTOF/MS) and identified in raw and extruded ML. The tentative identification and quantification are shown in Figure 2 and Table 3. Kaempferol and quercetin exist naturally as glycosides in MLs, and rutin and isoquercitrin are generally reported as main flavonols [3,10]. As shown in Figure 2, fifteen flavonols were identified from extruded ML, and quercetin-3-*O*-glucoside (Peak 9, isosquercitrin) was detected at the highest level (Table 3, 497.27 mg/ 100 g dry weight). Compounds with a content of 100 mg or more were in the order of isoquercitrin (Peak 9), quercetin-3-*O*-rutinoside (Peak 7, rutin), kaempferol-3-*O*-glucoside (Peak 13, astragalin), quercetin-3-gentiobioside (Peak 1), quercetin-3-*O*-(6”-*O*-malonyl)glucoside (Peak 10), and kaempferol-3-*O*-(6”-*O*-malonyl)glucoside (Peak 14) (Figure 2 and Table 3).

As a result, TF was significantly higher in extruded ML (1640.71 mg/100 g DW) compared with raw ML (1319.17 mg/100 g DW). In addition, extrusion induced an increase in nine flavonoids compared with raw ML. Among these, quercetin-3-gentiobioside (Peak 1), quercetin-3,7-di-*O*-glucoside (Peak 3), kaempferol-3,7-di-*O*-glucoside (Peak 5, morkotin B), rutin (Peak 7), isoquercitrin (Peak 9), and kaempferol-3-*O*-(6″-*O*-malonyl)glucoside-7-*O*-rhanmnoside (Peak 12, moragrol C) were significantly higher in extruded ML compared with raw ML.

### 2.3. Antioxidant Properties

#### 2.3.1. Radical Scavenging Activities of Extruded ML

In general, a good correlation has been reported between the antioxidant property (i.e., free radical scavenging activity) and the TP in plant samples [27]. The ability of radical scavenging is determined by hydrogen donation (1,1-diphenyl-2-picrylhydrazyl, DPPH) and electron donation as proton radical scavenging (2,2′-azino-bis-(3-ethylbenzthiazoline-6-sulfonic acid) diammonium salt, ABTS) [28]. In this study, the extruded ML had higher TP and TF than the raw ML (Table 4). Extruded ML water extract had a significantly lower the half maximal inhibitory concentration (IC_50_) value in both DPPH**^•^** and ABTS^•+^ scavenging activities than the raw ML, and it showed better inhibitory activity against DPPH^•^ than ABTS^•+^.

#### 2.3.2. Extruded ML Inhibits H_2_O_2_-Induced Apoptosis and Generation of ROS

Based on the antioxidant activity of extruded ML, the effects of ROS-mediated oxidative stress activities were studied in RAW 264.7 macrophages. As shown in Appendix A, cells treated with each ML water extract at a concentration of up to 500 μg/mL showed no effect on cell viability compared with the non-treatment control group (0 μg/mL). The protective effect against cytotoxicity stimulated by H_2_O_2_ was evaluated, and the results showed that ML water extracts significantly restored cell viability in a concentration-dependent manner (Figure 3a). In addition, Hoechst staining was carried out for the assessment of cytoprotection against apoptosis induction and nucleus morphology changes within the nuclear membrane of the cells treated with ML water extract and H_2_O_2_ stimulation (Figure 3b). The fluorescent images revealed that the normal cells had intact nuclei but that the H_2_O_2_-stimulated cells showed chromatin condensation (see arrows in Figure 3b). However, the morphological changes were attenuated in the cells pretreated with ML water extract before H_2_O_2_ stimulation.

Oxidative stress commonly leads to increased intracellular ROS levels, which has a fatal effect on oxygen toxicity and cellular function [29]. In the present study, intracellular ROS levels after ML water extract treatment were measured using the fluorescent probe 2′,7′-dichlorodihydrofluorescein diacetate (DCF-DA) (Figure 4). Pretreatment with each ML water extract decreased H_2_O_2_-stimulated ROS levels. In particular, treatment with extruded ML water extract resulted in a significantly lower level of ROS compared with raw ML water extract at a concentration of 500 μg/mL.

## 3. Discussion

The quality of the extruded product may vary depending on T, M, and S. The compromised optimal condition was determined for the development of extruded ML with improved TF. TF of extruded ML was significantly dependent on M, which had a negative linear effect. TF decreased with increasing T and M. Besides, M and S for TF showed that TF increased with decreasing M. In agreement with these results, previous studies indicated that radish leaves extruded at M of 20% had higher TF than those treated at higher M (25% and 30%) [30]. TP and its antioxidant activity of purple-flesh sweet potato flours extruded at M of 10% were higher than those treated at higher values of M (13% and 16%) [31]. The decrease of polyphenol compounds with an increase of M may be attributed to the fact that a large amount of moist heat is more destructive and has a synergistic affect along with higher T [20]. In addition, high M may promote polymerization of phenolics, resulting in reduced extractability in ML [23,32]. Similarly, higher S produces high shear to get high T during the procedure, so it has a negative effect on phenolic compounds [19]. Nevertheless, a notable prior study reported that appropriate heat processes for ML (i.e., steaming at 100 °C for 30 min, roasting at 200 °C for 5 min, microwave heating for 5 min) are effective for enhancing the level of TP [33]. These results explained that thermal processing might increase the extractable form of flavonoids in extrudates, and it could be attributed to the disruption of plant cell walls that provide better extractability and form soluble low-molecular-weight polyphenol compounds [34,35]. It is important to consider the optimal conditions affecting processing factors and the bioavailability of plant antioxidants for the utilization in food systems [32]. Hence, the optimal extrusion condition determined by a significantly fitted regression model for TF was T of 114 °C, M of 20%, and S of 232 rpm.

The extrusion process can enhance nutritional quality [13]. Under the optimal condition, the mean actual TF% (0.91 ± 0.04) was improved by 63% compared to the raw ML (0.56 ± 0.02). A previous study showed that extruded cranberry pomace was observed to have a significant increase (from 30% to 34%) in TF compared to the unextruded control [36]. Besides, the extrusion process elevated TP in sorghum bran (3.07 mg gallic acid equivalents GAE/g) compared to unextruded bran (2.02 mg GAE/g) [15]. A prior study observed that the processing of raw formulations using extrusion significantly raised the total oxygen radical absorbance capacity [34]. Flavonoids are well known to have multiple bioactivities, including antioxidant activity, and cell-protective effects [4]. Cellular antioxidant activity can be a biological means to quantify the bioactivity of antioxidants in food products because it considers the cellular uptake [34]. Interestingly, in the present study, the optimum extrusion process with MLs led to a significant increase in antioxidant properties, such as DPPH and ABTS radical scavenging following increasing TP and TF. Consistent with these results, inhibition of ROS generation in RAW 264.7 macrophage cells was statistically more effective by extruded ML than the raw ML. Extrusion induced an increase in the levels of quercetin derivatives morkotin B and moragrol C that were recently identified as flavonoids in ML and mulberry fruit [10,26]. In addition, applying the extrusion process to ML increased the levels of quercetin-3-gentiobioside, rutin, and isoquercitrin compared to the raw ML. These compounds were associated with important biological activities, such as antioxidant and anti-inflammatory effects [37]. In particular, a previous report indicated that quercetin-3-gentiobioside exhibited stronger antioxidative activities than α-tocopherol, and similar to that of rutin and butylated hydroxyanisole [24]. Previous studies have been conducted to optimize extraction methods for functional compounds in ML that are generally extracted by solvents [8,38]. However, the current study particularly has shown significant results of antioxidative properties as ML water extract. Higher antioxidant water extracts are more available than solvent extracts for use at a commercial level.

Taken together, the findings of this study suggest that the extrusion process can be an alternative food processing strategy for MLs, and its optimization may improve functional compounds and antioxidant activity, beneficial for health. In addition, the availability of MLs with higher contents of flavonoids as value-added food materials could be a potential base for the manufacturing of food products.

## 4. Materials and Methods

### 4.1. Materials

Dried mulberry leaf powder (Daeshim) was obtained from the Sericulture and Apiculture Division of the National Institute of Agriculture Sciences, Rural Development Administration, South Korea, in 2019. The following materials were obtained from Sigma-Aldrich Co. (St. Louis, MO, USA): Folin-Ciocalteu (F-C) reagent, sodium carbonate (Na_2_CO_3_), gallic acid, aluminum trichloride (AlCl_3_), potassium acetate (CH_3_COOK), quercetin, 1,1-diphenyl-2-picrylhydrazyl (DPPH**^•^**), potassium persulfate (K_2_S_2_O_8_), 2,2′-azino-bis-(3-ethylbenzthiazoline-6-sulfonic acid) diammonium salt (ABTS), kaempferol, hydrogen peroxide (H_2_O_2_), 2′,7′-dichlorodihydrofluorescein diacetate (DCF-DA), and Hoechst 33,342. Acetonitrile (ACN), methanol (MeOH), and purified water were purchased from Fisher Scientific (Fair Lawn, NJ, USA). Dulbecco’s modified Eagle’s medium (DMEM) and penicillin-streptomycin were purchased from HyClone (Logan, UT, USA). Fetal bovine serum (FBS) was obtained from Grand Island Biochemical Co. (Grand Island, NY, USA). EZ-Cytox was supplied by DoGenBio (Seoul, Korea).

### 4.2. Response Surface Modeling

To determine the levels of the extrusion condition, the effect of extrusion on ML was examined by varying two extrusion factors: T (110, 120, and 130 °C) and M (20% and 30%). S was constant at 250 rpm. The analysis of the main effect and the interaction plot were based on factorial design performed using Minitab statistical software (version 19; Minitab, Ltd., State College, PA, USA). Then, to optimize the extrusion process using RSM, T, M, and S were set as independent variables (i.e., T, 110–130 °C; M, 20–40%; S, 200–300 rpm). The levels were coded using a three-level and three-factor factorial Box-Behnken design, which yielded 15 experimental variations (Table 1). The response variables were fitted with a quadratic polynomial equation:Y_ijk_ = β_0_ + β_1_X_1i_ + β_2_X_2j_ + β_3_X_3k_ + β_11_X_1_^2^_i_ + β_22_X_2_^2^_j_ + β_33_X_3_^2^_k_ + β_12_X_1i_X_2j_ + β_13_X_1i_X_3k_ + β_23_X_2j_X_3k_(4)
where Y was the response variable for the level of X_1_ (T), X_2_ (M), and X_3_ (S). Analysis of variance (ANOVA) was performed to evaluate significant differences and check the adjusted and predicted coefficient of determination (R^2^) values.

### 4.3. Extrusion Process

The extruder was a laboratory-scale, co-rotating, intermeshing twin-screw extruder (THK31T; Incheon Machinery Co., Incheon, Korea). The circular die diameter was 3.0 mm, and the ratio of screw length to screw diameter was 23:1. The screw configurations of the extruder are given in Appendix A. The extrusion was carried out at each designed T, M, and S, as shown in Table 1. The feed rate for all samples was 100 g/min. The final optimal extrusion process was conducted at 114 °C, 20%, and 232 rpm. During extrusion, the desired M was controlled by a water feed pump (Grundfos DDA 30-4; Grundfos GmbH, Pfinztal, Germany) applied directly into the sample feeding zone.

### 4.4. Sample Collection and Preparation

The extrudates were collected at each condition and held at room temperature (RT) in an area of subdued light. Drying was done in an air oven (OF-22; Jeio Tech Co., Ltd., Daejeon, Korea) at 50 °C for 24 h. The dried extrudates were ground to a powder using a stainless-steel blender (SMX-C4000WK; Hanil Electronics Co., Ltd., Incheon, Korea), sieved (35 mesh, 500 μm), and kept in closed bags at −20 °C until analysis. The moisture content (wt%) of the extrudates was determined (2.0–5.4 wt%). For extraction, 5.0 g of ground extrudate was mixed in 100 mL of distilled water at 100 °C for 2 h, and the filtered water extract of the optimal extrudates was freeze-dried (Bondiro freeze dryer; IlShinBioBase Co., Ltd., Seoul, Korea) for antioxidant activity analysis.

### 4.5. Water Solubility Index (WSI)

WSI was determined using the method of Anderson [39] with some modifications. Briefly, 5.0 g of ground extrudate was extracted in 100 mL of distilled water at 100 °C for 2 h and then centrifuged at 2500× *g* for 15 min. Next, 5 mL of the supernatant was transferred into an evaporating dish. The supernatant was dried to constant weight in an oven at 50 °C for 24 h.
%WSI = dried supernatant (g)/supernatant (5 mL) × total supernatant (mL)/sample (g dryweight) × 100(5)

### 4.6. Total Phenolic (TP) Content

TP was examined using the F–C method, as reported by Ainsworth and Gillespie [40], with slight modifications. Briefly, 0.1 mL of 5-fold diluted extract was mixed with 0.2 mL of 10% F–C reagent and kept at RT for 5 min. Then, 0.8 mL of 700 mM Na_2_CO_3_ was added to each mixture and kept at RT for 30 min. Subsequently, 0.2 mL of the mixture was transferred into a 96-well plate, and the absorbance was determined at 765 nm (Infinite M200 Pro; Tecan, Krems, Austria). A standard curve was used in order to estimate phenolics (gallic acid equivalents, GAE).
%TP = gallic acid (g/mL) × total extract (mL)/sample (g dry weight) × 100(6)

### 4.7. Total Flavonoid (TF) Content

TF was determined by the AlCl_3_ method [41]. A volume of 0.1 mL of 5-fold-diluted extract was mixed with 0.005 mL of 10% AlCl_3_ and 0.005 mL of 1 M CH_3_COOK in a 96-well plate. The final mixture was adjusted to 0.25 mL with distilled water and kept at RT for 30 min. The absorbance was read at 415 nm (Infinite M200 Pro; Tecan). A standard curve of quercetin was used to measure the TF of samples (quercetin equivalents, QE).
%TF = quercetin (g/mL) × total extract (mL)/sample (g dry weight) × 100(7)

### 4.8. UPLC-PDA-QTOF/MS Analysis

Freeze-dried sample (0.3 g) was suspended in 30 mL of MeOH:water:formic acid extraction solution (50:45:5, *v*/*v*/*v*) [26]. The extract was purified by solid-phase extraction using a Sep-Pak C_18_ cartridge (Waters Co., Milford, MA, USA). The analysis was performed on an ultra-high-performance liquid chromatograph coupled with photodiode array detection (350 nm) and quadrupole time-of-flight mass spectrometry (UPLC-PDA-QTOF/MS; Waters Co.). Ultraviolet spectra were taken in the range of 210–400 nm. The chromatographic conditions and equipment were as follows: Kinetex C_18_ column (XB-C18 100A, 150 × 2.1 mm inner diameter (i.d.), 1.7 μm; Phenomenex, Torrance, CA, USA), pre-column (Acquity UPLC BEH C18, 2.1 mm i.d., 1.7 μm; Waters Co.), injection volume 1 μL, column temperature 30 °C, and running time, 40 min. Flavonoids were separated at a flow rate of 0.3 mL/min with mobile phases of 0.5% formic acid in water (A) and 0.5% formic acid in ACN (B). The following gradient conditions were used: 5% B (initial), 25% B (20 min), 50% B (25 min), 90% B (30–32 min), and 5% B (35–40 min). Mass spectra were recorded in the range of m/z 100–1200 with electrospray ionization in positive ion mode (source temperature, 120 °C; desolvation temperature, 500 °C; desolvation N_2_ gas flow, 1050 L/h; cone gas, 50 L/h; capillary voltage, 3500 V; sampling cone voltage, 40 V). Data acquisition was performed with MassLynx v4.1 software (Waters Co.). Levels of compounds were determined by standard calibration curves (quercetin, y = 53.211x − 132.62, R^2^ = 0.9995; kaempferol, y = 25.14x − 51.751, R^2^ = 0.9993).

### 4.9. Antioxidant Properties

#### 4.9.1. Radical Scavenging Activity

Radical scavenging activities were evaluated using the DPPH^•^ and ABTS^•+^ assays [28]. Briefly, 0.15 mM DPPH solution was prepared in ethanol, and 0.16 mL of this solution was added to 0.04 mL of extract. The mixture was kept at RT for 30 min in the dark and the absorbance was recorded at 517 nm (Infinite M200 Pro; Tecan). For the production of ABTS^•+^, 7.4 mM ABTS was reacted with 2.6 mM K2S2O8 and stored in the dark at RT for 12 h. The ABTS^•+^ solution was diluted to yield an absorbance of 1.2–1.5 at 734 nm. Then, 0.015 mL of extracts were placed in a 96-well plate, 0.285 mL of the ABTS^•+^ solution was added, and the absorbance was measured at 734 nm after 5 min.
%Radical scavenging = [(absorbance of control − absorbance of sample)/absorbance ofcontrol] × 100(8)

Then, curves between the percentage of scavenging and each extract concentration in mg/mL were plotted, and the half-maximal inhibitory concentration (IC_50_) value was determined.

#### 4.9.2. Intracellular ROS Production

The level of intracellular ROS was determined using DCF-DA dye [42]. Mouse macrophage cells, RAW 264.7 (TIB-71; American Type Culture Collection, Rockefeller, MD, USA), were maintained in DMEM containing 10% FBS, 100 U/mL penicillin, and 0.1 mg/mL streptomycin, and cells were seeded in a 96-well black plate for measurement of fluorescence intensity or a 24-well plate for microscope observation. The cells were preincubated with extract for 2 h before H_2_O_2_ treatment for another 1 h. After that, cells were added to DCF-DA dye (0.01 mM) and incubated for 30 min at 37 °C. Fluorescence intensity was measured using a plate reader (Infinite F200 Pro; Tecan) at an excitation wavelength of 485 nm and emission wavelength of 528 nm, and images were visualized using a fluorescence microscope (DMi8; Leica Microsystems, Wetzlar, Germany).

#### 4.9.3. Apoptosis Assay

To observe the nuclear morphological changes in apoptotic cells, cells were seeded in a 6-well plate [43]. The cells were pretreated with extract for 2 h and cultured in the presence or absence of H_2_O_2_ for 24 h. Then, cells were treated with Hoechst 33,342 staining at 1 mg/mL for 30 min and washed twice with PBS. The morphology changes in the nucleus were observed using a fluorescence microscope (DMi8; Leica Microsystems). Cell viability was determined using the EZ-Cytox assay kit [28].

### 4.10. Statistical Analysis

All of the data in this study were analyzed using SPSS software (version 20.0; IBM Corp., Armonk, NY, USA). Comparisons were made using ANOVA with Duncan’s multiple range tests and independence tests. A *p*-value < 0.05 was considered to indicate statistical significance.

## Figures and Tables

**Figure 1 molecules-25-05231-f001:**
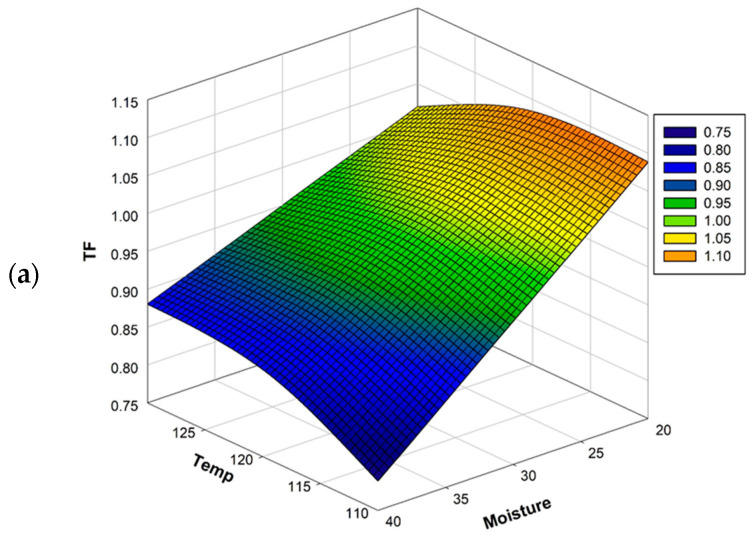
Response surface plots showing (**a**) the effects of temperature and moisture content, (**b**) the effects of screw speed and moisture content, and (**c**) the effects of screw speed and temperature on total flavonoid content. Temp, barrel temperature (°C); Moisture, feed moisture content (%); Speed, screw speed (rpm). TF, total flavonoid content (%).

**Figure 2 molecules-25-05231-f002:**
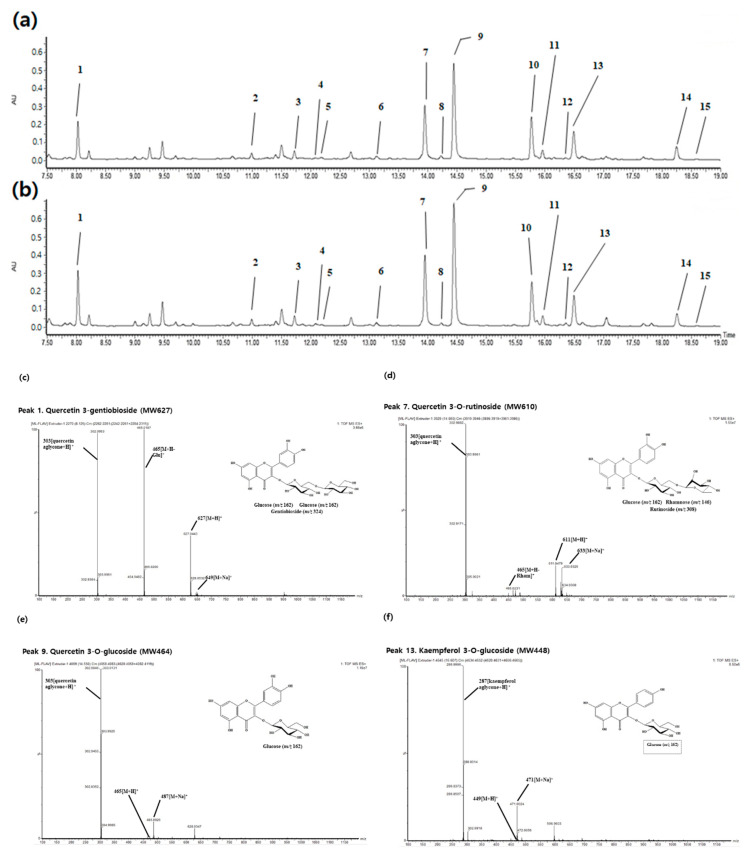
Liquid chromatograms of flavonoids in (**a**) raw mulberry leaves and (**b**) the optimal extruded mulberry leaves (114 °C, 20%, 232 rpm). (**c**–**f**) Mass spectrum of major flavonoids. (**c**) Quercetin 3-gentiobioside, (**d**) Quercetin 3-*O*-rutinoside, (**e**) Quercetin 3-*O*-glucoside, and (**f**) Kaempferol 3-*O*-glucoside. The names of the flavonoids are listed in Table 3.

**Figure 3 molecules-25-05231-f003:**
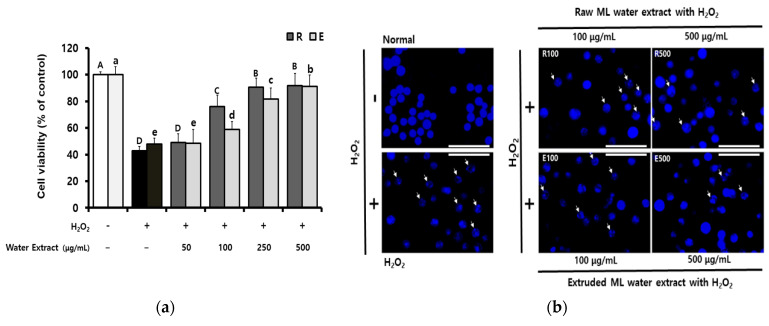
Protective effect of extruded mulberry leaves water extract against H_2_O_2_-induced oxidative damage in RAW 264.7 cells. (**a**) Cells were pretreated with or without extract for 2 h and then cultured with H_2_O_2_ for 24 h. (**b**) The cells were stained with Hoechst 33,342 solution and nuclei were observed using a fluorescence microscope (magnification, 400×; scale bar, 50 μm). ML, mulberry leaves; R, raw ML water extract; E, extruded ML water extract. Data are expressed as the means ± SD. Different uppercase letters for (R) or lowercase letters for (E) represent significant differences between groups according to analysis of variance (ANOVA) followed by Duncan’s multiple range test (*p* < 0.05).

**Figure 4 molecules-25-05231-f004:**
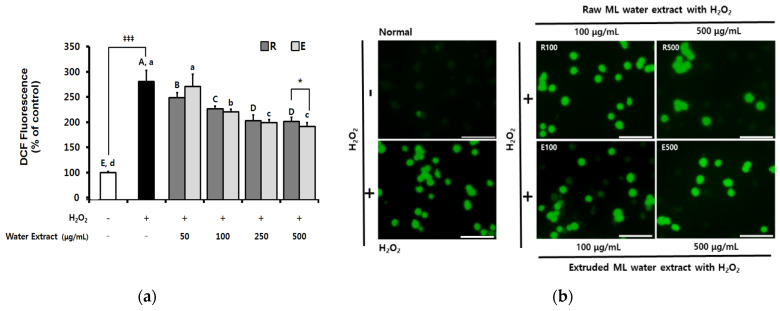
Inhibition of H_2_O_2_-induced reactive oxygen species (ROS) generation by extruded mulberry leaves’ water extract in RAW 264.7 cells. (**a**) The degree of ROS production as measured by dichlorofluorescein fluorescence and (**b**) images obtained by fluorescence microscopy (magnification, 200×; scale bar, 100 μm). ML, mulberry leaves; R, raw ML water extract; E, extruded ML water extract. Data are expressed as the means ± SD. Different uppercase letters for (R) or lowercase letters for (E) represent significant differences between groups according to ANOVA followed by Duncan’s multiple range test (*p* < 0.05). H_2_O_2_-stimulated group without sample treatment compared with unstimulated control group by independent *t*-test (^‡‡‡^
*p* < 0.001). R group compared with E group by independent *t*-test (* *p* < 0.05).

**Table 2 molecules-25-05231-t002:** Regression summaries for independent variables.

Parameters	Regression Parameter Coefficients		
	WSI (%)	TP (%)	TF (%)
T (X_1_)	0.195	0.0009	0.0067
M (X_2_)	−1.038 **	−0.0590 *	−0.0941 ***
S (X_3_)	0.439	−0.0215	0.0168
T × M (X_1_X_2_)	0.487	0.0167	0.0401 *
T × S (X_1_X_3_)	.	−0.0312	.
M × S (X_2_X_3_)	.	0.0173	0.0441 **
T × T (X_1_^2^)	0.256	.	−0.0304 *
M × M (X_2_^2^)	0.697	.	.
S × S (X_3_^2^)	−0.396	−0.0355	−0.0395 *
F-value (model)	3.38	2.08	25.19 ***
Lack of fit	NS	NS	NS
R^2^	0.772	0.676	0.962
Adjusted R^2^	0.544	0.351	0.924
Predicted R^2^	0	0	0.773

T, barrel temperature (°C); M, feed moisture content (%); S, screw speed (rpm). WSI, water solubility index; TP, total phenolic content; TF, total flavonoid content. Asterisks indicate significant difference (* *p* < 0.05; ** *p* < 0.01; *** *p* < 0.001). NS, not significant.

**Table 3 molecules-25-05231-t003:** Contents of flavonoids (mg 100 g/dry weight) isolated from optimal extruded mulberry leaves extract.

Peak No.	Compound	MW	Fragment Ions (*m*/*z*)	Raw	Extrusion	Reference
1	Quercetin 3-gentiobioside	627	649, 627, 465, 303	95.94 ± 8.05	152.75 ± 4.99 ***	[24,25]
2	Quercetin 3-*O*-rutinoside-7-*O*-glucoside (morkotin A)	772	795, 773, 627, 611, 465, 303	15.79 ± 1.19 ***	2.81 ± 0.05	[3,9,26]
3	Quercetin 3,7-di-*O*-glucoside	626	649, 627, 465, 303	44.84 ± 2.84	57.06 ± 5.17 *	[3,9,26]
4	Kaempferol 3-*O*-rutinoside-7-*O*-glucoside (moragrol A)	756	779, 757, 611, 595, 449, 287	3.84 ± 0.34 **	2.48 ± 0.11	[9]
5	Quercetin 3-*O*-rutinoside-7-*O*-rhamnoside (morkotin B)	756	779, 757, 611, 465, 449, 303	9.96 ± 1.32	19.71 ± 0.67 ***	[9]
6	Kaempferol 3-*O*-rutinoside-7-*O*-rhamnoside (moragrol B)	740	763, 741, 595, 449, 433, 287	22.13 ± 1.05	22.71 ± 0.90	[9]
7	Quercetin 3-*O*-rutinoside (rutin)	610	633, 611, 465, 449, 303	196.79 ± 24.06	267.32 ± 12.39 *	[3,9,26]
8	Kaempferol 3-*O*-rhamnoside-7-*O*-glucoside	594	617, 595, 449, 433, 287	21.56 ± 2.12	19.75 ± 0.68	[9]
9	Quercetin 3-*O*-glucoside (isoquercitrin)	464	487, 465, 303	368.65 ± 48.46	497.27 ± 20.32 *	[3,9,26]
10	Quercetin 3-*O*-(6″-*O*-malonyl)glucoside	550	573, 551, 465, 303	137.93 ± 20.95	147.39 ± 17.40	[3,9,26]
11	Kaempferol 3-*O*-rutinoside (nicotiflorin)	594	617, 594, 447, 287	53.46 ± 4.82	58.89 ± 2.70	[3,9,26]
12	Kaempferol 3-O-(6’’-*O*-malonyl)glucoside-7-*O*-rhamnoside (moragrol C)	680	703, 681, 535, 433, 287	9.27 ± 1.07	17.78 ± 0.79 ***	[9]
13	Kaempferol 3-*O*-glucoside (astragalin)	448	471, 449, 287	225.64 ± 23.23	266.02 ± 10.79	[3,9,26]
14	Kaempferol 3-*O*-(6″-*O*-malonyl)glucoside	534	557, 535, 449, 287	105.81 ± 9.71	102.07 ± 11.77	[3,9,26]
15	Kaempferol 3-O-(2″-*O*-malonyl)glucoside (moragrol D)	534	557, 535, 449, 287	7.55 ± 0.65	6.70 ± 00.71	[9]
	Quercetin-based flavonoid			859.95 ± 102.59	1124.60 ± 54.57 *	
	Kaempferol-based flavonoid			459.23 ± 43.28	516.11 ± 26.94	
	Total flavonoid			1319.17 ± 145.87	1640.71 ± 81.25 *	

All samples were analyzed in positive ion mode (*m*/*z*, [M + H]^+^) using ultra-high-performance liquid chromatograph coupled with photodiode array detection and quadrupole time-of-flight mass spectrometry. Each value is expressed as the mean ± standard deviation (SD). Raw group compared with extrusion group by independent *t*-test (* *p* < 0.05, ** *p* < 0.01, *** *p* < 0.001). MW, molecular weight.

**Table 4 molecules-25-05231-t004:** Total phenolic content, total flavonoid content, and radical scavenging IC_50_ values of freeze-dried water extract from optimal extruded mulberry leaves.

	TP (mg GAE/g FDW)	TF (mg QE/g FDW)	DPPH^•^ IC_50_ (mg/mL)	ABTS^•+^ IC_50_ (mg/mL)
Raw	38.94 ± 2.06	37.87 ± 0.59	7.09 ± 0.91	5.85 ± 0.03
Extrusion	51.43 ± 1.11 *	43.75 ± 0.78 *	2.95 ± 0.66 *	4.47 ± 0.20 *

TP, total phenol content; TF, total flavonoid content, GAE, gallic acid equivalents; QE, quercetin equivalents; FDW, freeze-dried weight; DPPH^•^, 1,1-diphenyl-2-picrylhydrazyl; ABTS^•+^, 2,2′-azino-bis-(3-ethylbenzthiazoline-6-sulfonic acid) diammonium salt; IC_50_, the half maximal inhibitory concentration. Each value is expressed as the mean ± SD. Raw group compared with extrusion group by independent *t*-test (* *p* < 0.05).

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
