# Peer review of "Response Surface Methodology for Optimization of Process Parameters and Antioxidant Properties of Mulberry (*Morus alba* L.) Leaves by Extrusion"

_molecules, 2020, doi:10.3390/molecules25225231_

Round 1

Reviewer 1 Report

In this manuscript the authors reported the extrusion process applied for the enhancement of flavonoids content from ML.

Below are my comments:

1) Although the authors claimed that the TF of extruded ML reached to 0.91% and improved by 63% compared with raw ML. It would be better if the authors also compare the extrusion process to other traditional process such as reflux or sonication, to tell whether the extrusion process is better than all other processes. 

2) I am not quite clear what is the "different superscripts" in the table 1 meaning. 

3) During the determination of the "Contents of flavonoids" in the table 3, Did the authors have the standard of all these metabolites? It seems the authors only have standard calibration curves of quercetin, and kaempferol.

4) During the extraction protocol, the authors used distilled water at 100 °C for 2 h, Why the authors did not use the organic solvents as the extracting solvent, such as MeOH-H2O or EtOh-H2O, these solvents have much better solubility for most of the metabolites. 

Reviewer 2 Report

General

This study optimized extrusion parameters for mulberry leaves, determined the content of flavonoids in the extruded products, evaluated their radical scavenging capacity and conducted some assays in the cells. The manuscript is based on a large pool of data, which was generated by using reliable methods. However, to my opinion, the interpretation of the results is not correct and in many cases speculative, particularly in using the statements regarding increase/decrease of bioactive compounds. The following points should be considered for avoiding this problem: (1) the compounds cannot be ‘synthesized’ (‘increase’) during extrusion (some sensitive to heat compounds may even degrade); (2) flavonoids are present in glycosidic form (Table 3) and the glycosides may (at least partially) undergo hydrolysis during extrusion and loose sugar moiety(ies); however, in this case aglycones should form, while they are not present in Table 3; (3) some flavonoids may be strongly bound in the polysaccharide matrix of leaves and not fully recovered from the raw material before extrusion. Consequently, to my opinion, discussion should be rewritten by considering these points and avoiding speculative interpretations. E.g. instead of increased amount of TP, TF, etc. the terms ‘the increased amount of extractable substances, TP, TF’, etc. should be used. In this case, comparisons with the previously reported data should be also selected more carefully: published data does not mean that it is absolutely correct. Therefore, published results, which are selected for citations, should be assessed critically. Some specific remarks follow.

Specific remarks

Title: ‘enriched’ is not a proper word for indicating that the leaves containing flavonoids; the title should be revised b selecting adequate terms.

Abstract

L.17: ‘medicinal substances’ officially mean pharmaceuticals; such substances are generally called ‘bioactive’, health beneficial’.

L.18: This is rather ambiguous statement.

L.22: 20% of what?

Introduction

L.34: Systematic name of silkworm is not required for this type of study.

L.37: Are they consumed for food as fresh or dried leaves?

L.41: I think, that the term ‘active hydrogen’ is not correct. These compounds just may donate a proton to radical species. The whole molecule in this case acts as a scavenger.

L.43: ‘abandoned’ = discarded

L.46-49: These sentences should be rephrased. In general the quality of language should be substantially improved (from this point grammar issues are not included into the review).

L.64-66: The task should be reformulated; extrusion itself cannot change (increase) the composition of bioactives in MLs (may be some more sensitive may be even degrade?).

Results

L.8: How extrusion can increase these variable? They cannot be synthesized during extrusion. The terms which adequately reflect these findings should be used, e.g. accessibility (in the F-C, TF assays), extractability. 

Table 4: It is a pity that more relevant to biological processes assays, e.g. ORAC, have not been applied. ORAC values could be better linked to the assays in the cells.

Discussion

L.199: If MLs are a good source of anthocyanins, why these compounds were not analyzed (detected)?

  1. 199-216: In general, these paragraphs duplicate the information provided in the Introduction.

Materials and methods

L.272: DPPH is a stable radical and it should be indicated by the word or better by the superscript dot throughout the text.

Reviewer 3 Report

The present study is very interesting, especially due to  the  extrusion process applied for the enhancement of flavonoids content from Morus alba, a plant species which was not studied before for its antioxidant activity.. In spite of the manuscript is pretty good written some improvement of it should be done.

Line 37   Recently, mulberry leaves (MLs) have been recognized as an edible food (edible ???? please check)

Line 51   research significance (you could change significance to interest)

Line 53    The industrial process of extrusion is a continuous, stepwise, and short-time thermomechanical operation using high pressure and shear force (Check phrase, especially word shear)

Line 58    delete and improving their biological….

Line 61    (add reference or references)

Line 63    delete employ  and replace with another verb

Line 87     what is screw speed?......

Explain independent variables, T and M within the text

Line 96   Replace R2 to R2  , Check the whole text

In general, do not leave a gap between refernec number i.e. (line 135… [3,25])

Line 166    in Figure. S4,    delete .

Line 322    standard curve was used in order to estimate phenolics

Please check again reference style. In some references the first letter of each word is capitalized while in other is not. i.e. references 1 and 2

Reference 14 …..2016 is duplicated

Round 2

Reviewer 2 Report

The manuscript has been revised by addressing Reviewers remarks and comments. However, language quality should be improved.

Reviewer 3 Report

It is obvious that the authors made a big effort to improve the previous manuscript, much more than I asked for, and now is much better.